# Development of High-Performance Thermoelectric Materials by Microstructure Control of P-Type BiSbTe Based Alloys Fabricated by Water Atomization

**DOI:** 10.3390/ma14174870

**Published:** 2021-08-27

**Authors:** Babu Madavali, Pathan Sharief, Kyoung-Tae Park, Gian Song, Song-Yi Back, Jong-Soo Rhyee, Soon-Jik Hong

**Affiliations:** 1Division of Advanced Materials Engineering, Institute for Rare Metals and Center for Advanced Powder Materials and Parts, Kongju National University, 275, Budae-dong, Cheonan City 31080, Chungcheongnam-do, Korea; mbabuphy@gmail.com (B.M.); shareefpathan2012@gmail.com (P.S.); gasong@kongju.ac.kr (G.S.); 2Rare metal R&D Group, Korea Institute of Industrial Technology, Incheon 31056, Korea; ktpark@kitech.re.kr; 3Department of Applied Physics, Integrated Education Institute for Frontier Science and Technology (BK21 Four) and Institute of Natural Sciences, Kyung Hee University, Yongin 17104, Korea; song2back@gmail.com (S.-Y.B.); jsrhyee@khu.ac.kr (J.-S.R.)

**Keywords:** water atomization, BiSbTe alloys, microstructure, hot pressing, thermoelectric properties

## Abstract

Developing inexpensive and rapid fabrication methods for high efficiency thermoelectric alloys is a crucial challenge for the thermoelectric industry, especially for energy conversion applications. Here, we fabricated large amounts of p-type Cu_0.07_Bi_0.5_Sb_1.5_Te_3_ alloys, using water atomization to control its microstructure and improve thermoelectric performance by optimizing its initial powder size. All the water atomized powders were sieved with different aperture sizes, of 32–75 μm, 75–125 μm, 125–200 μm, and <200 μm, and subsequently consolidated using hot pressing at 490 °C. The grain sizes were found to increase with increasing powder particle size, which also increased carrier mobility due to improved carrier transport. The maximum electrical conductivity of 1457.33 Ω^−1^ cm^−1^ was obtained for the 125–200 μm samples due to their large grain sizes and subsequent high mobility. The Seebeck coefficient slightly increased with decreasing particle size due to scattering of carriers at fine grain boundaries. The higher power factor values of 4.20, 4.22 × 10^−3^ W/mk^2^ were, respectively, obtained for large powder specimens, such as 125–200 μm and 75–125 μm, due to their higher electrical conductivity. In addition, thermal conductivity increased with increasing particle size due to the improvement in carriers and phonons transport. The 75–125 μm powder specimen exhibited a relatively high thermoelectric figure of merit, ZT of 1.257 due to this higher electric conductivity.

## 1. Introduction

Thermoelectric renewable energy devices can operate in two directions, in power generation mode and thermoelectric cooling mode. The power generation mode is based on the Seebeck effect, where a temperature gradient between two different materials can develop a voltage. On the other hand, thermoelectric cooling is based on the Peltier effect, where cooling and heat can be generated at the junctions of two different materials by maintaining an electrical current [1,2,3]. Generally, the quality of a thermoelectric device is evaluated using the dimensionless figure of merit, ZT = σα^2^T/κ, where *σ* is electrical conductivity in 1/Ωcm, *α* is the Seebeck coefficient in μV/K, *T* is the working temperature in K, and *κ* is thermal conductivity in W/m-K [4,5,6]. Materials that have a high Seebeck coefficient and low thermal conductivity, such as ceramics/glass materials, and simultaneously exhibit high electrical conductivity, such as single crystals, can exhibit high thermoelectric properties [4]. Many researchers have developed various thermoelectric materials, including Bi-Te based alloys [7,8], Pb-Te based alloys [9], clathrates (Ba-Cu-Si) [10], skutterudites (Co_4_Sb_12_) [11], Ag-Pb based alloys [12], and half Heusler [13] in efforts to enhance thermoelectric performance.

Among them, the binary and ternary compounds of bismuth-telluride (Bi-Te) based alloys exhibit excellent room temperature thermoelectric performance due to their unique properties, including a narrow band gap and ease of doping with metallic/semiconductor elements [7,8]. In the last few years, investigators have focused on various novel approaches/methods to improve the ZT values above unity (ZT > 1) by controlling the materials’ microstructure using nanostructuring approaches, such as the nano-dispersion of ceramics/semiconductor elements and doping via powder metallurgical processes [14,15,16,17,18]. Poudel et al. reported p-type Bi-Sb-Te based nano-crystalline (NC) specimens with a maximum ZT of 1.4 at 100 °C by ball milling and hot pressing [14]. They reported that nanostructured grains were formed during the hot pressing, which were responsible for a significant reduction in total thermal conductivity due to the severe scattering of phonons at the interfaces. Kim et al. developed dense dislocation arrays in p-type Bi_0.5_Sb_1.5_Te_3+x_ alloys at the grain boundaries, and achieved a ZT of 1.8 at 320 K [15]. They argued that substantial low and high frequency phonon scattering occurred at the dense dislocations, significantly enhancing the thermoelectric properties. Recently, Back et al. argued that 0.03 wt% Ga excess BiSbTe alloys exhibited a high ZT of 1.13 due to a significant decrease in lattice thermal conductivity owing to phonon scattering [16] at the interfaces. In addition to microstructure modifications, suppressing intrinsic conduction at high temperature is crucial in BiSbTe alloys, as the bipolar conduction effect can detrimentally affect thermoelectric performance. In this instance, approaches that widen the band gap and/or increase carrier concentration can strongly suppress the bipolar conduction. Hu et al. investigated the Sb-doping effect on BiSbTe alloys, and concluded that the increase in Sb content was responsible for the increase in hole concentration as well as the alloy band gap. The intrinsic conduction was dramatically reduced at high temperatures, and achieved a maximum ZT of 1.3 at 380 K [17]. Cao et al. also studied the suppression of intrinsic conduction in Ag doped BiSbTe alloys and achieved a high ZT of 1.07 in a 0.03 wt% Ag specimen due to high electrical conductivity, thanks to their high carrier concentration values [18].

Despite these improvements in thermoelectric and mechanical properties, the ascribed powder metallurgical processes have low production rates and are expensive and time-consuming methods. In comparison, the preparation of thermoelectric alloys using gas atomization (GA) process is an excellent alternative. GA can be used to fabricate desired powders at a large scale through a rapid solidification process. It can produce 2–3 kgs/min of desired powders with a homogeneous chemical composition and fine-grained alloys [19]. Recently, we prepared p-type 25%Bi_2_Te_3_ + 75%Sb_2_Te_3_ alloys using GA followed by a spark plasma sintering process, and investigated the influence of powder size on thermoelectric properties. In this report, the grain size increased with increasing powder size and, as a result, electrical conductivity was increased, due to increasing carrier mobility values. We obtained a maximum ZT of 1.2 with a 32–75 μm specimen due to the high power factor and reasonable thermal conductivity [20]. However, it is necessary to maintain an inert atmosphere in the gas atomization process, and the use of nitrogen and argon gasses is expensive. To overcome these limitations, we have designed an atomization process using water injection (water atomization, WA) instead of gas, and succeeded in synthesizing powders.

In the WA process, it is not necessary to maintain a vacuum or have inert gas in the chamber, as in the GA process, which significantly reduces production time and cost. Hot pressing is a traditional sintering method used to make Bi_2_Te_3_-based pellets due to its simple procedure, and produces a high density [21]. Cu is chosen as a dopant in the BiSbTe system to suppress intrinsic conduction at elevated temperatures. In this study Cu_0.07_Bi_0.5_Sb_1.5_Te_3_ alloys were fabricated in large-scale quantity using water atomization followed by hot pressing, rather than the un-doped BiSbTe and SPS in the previous work. In the present work, we have mainly focused on the effect of different WA particle sizes on microstructural features, and their subsequent effect on thermoelectric properties and mechanical properties were systematically investigated. In addition, the suppression of intrinsic/bipolar conduction due to the Cu content in vBi_0.5_Sb_1.5_Te_3_ alloys was also explored at high temperatures.

## 2. Experimental Procedure

High purity (5N) of Bi, Sb, Cu, and Te were used (purchased from Alfa Aesar Co., Seoul, South Korea) according to the stoichiometric portions of the Cu_0.07_Bi_0.5_Sb_1.5_Te_3_ alloys. The atomization chamber was initially filled with up to 60% water instead of inert gas. All of the elements were placed into a graphite crucible inside an induction furnace and maintained 100 °C above the melting point, as monitored by a thermocouple. Later, the melt solution was poured into a boron nitrate-based nozzle set up, of 5 mm in diameter, and disintegrated with the water stream at 40 bar pressure. The fabricated wet powder was collected and dried in a vacuum chamber at 180 °C for 8 h. The fabricated powders had a wide range of particle sizes due to the different in cooling rates during the process. A high cooling rate resulted in a fine powder size, while the slow cooling rate resulted in coarse powders [20,22]. To investigate the impact of powder size on microstructural features, and mechanical and thermoelectric properties, mechanical sieving was used, and the powders were classified as 32–75 μm, 75–125 μm, 125–200 μm, and <200 μm. All of the powders were sintered using hot pressing at 490 °C for 30 min under 50 MPa.

The powder morphology, microstructure and bulk fracture surface analysis were characterized using scanning electron microscopy (SEM- MIRA-LMH II (TESKAN)). The particle size was measured using laser diffraction particle size analyzer (SALD-2300, Shimadzu). The density (d), was measured 10 times using the Archimedes principal, and we averaged its value. Micro-Vickers hardness was obtained using a Vickers indenter measurement system with an indenting load of 300 gF. The oxygen contamination was analyzed using an Oxygen Nitrogen Hydrogen—2000 analyzer (ELTRA GmbH). The crystal structures of the powders and bulk samples were examined using an X-ray diffractometer at room temperature (Rigaku, MiniFlex-600). The hot-pressed samples were cut into small rectangular pieces about (4 × 4 × 12 mm^3^) for Seebeck coefficient and electrical conductivity, and into square pieces (10 × 10 mm^2^, 2 mm thickness) for measurement of thermal conductivity. The electrical conductivity and Seebeck coefficient were measured by thermoelectric power measurement system (TEP-1000, Seepel) from room temperature to 500 K. The thermal diffusivity (D), and specific heat (C_p_) were measured using the laser flash method (Netzsch-LFA-467 system) and a differential scanning calorimeter (PerkinElmer DSC8000). The Hall mobility and carrier concentrator were measured at room temperature using a four probe-hall system (Ecopia HMS-3000). The thermoelectric figure of merit, ZT was calculated using the Seebeck coefficient, electrical conductivity, and thermal conductivity (D × C_p_ × d). All thermoelectric transport properties were measured in the direction perpendicular to the pressing.

## 3. Results and Discussion

Figure 1 depicts the typical SEM morphology of the water atomized p-type Cu_0.07_Bi_0.5_Sb_1.5_Te_3_ alloy powder, in different size ranges of (a) 32–75 μm, (b) 75–125 μm, (c) 125–200 μm, and (d) <200 μm. The insets in (a)–(d) show their corresponding high magnification images. All the water atomized (WA) powder particles exhibited irregular/polygonal shapes. The rapid cooling rate due to the water stream during the atomization process was responsible for the formation of the irregular shapes, unlike the spherical powders produced by gas atomizer. The powders are clearly distinguishable according to their powder size, and the highly magnified images revealed that the water atomized powder were highly compacted. Figure 1e–h shows the cross-sectional microstructures of the (e) 32–75 μm, (f) 75–125 μm, (g) 125–200 μm, and (h) <200 μm powders. The cross-sectional microstructure images show an evenly distributed acicular grain on the surface [20]. In addition, the grain sizes gradually increased with increasing powder size. The fine powders contain fine grains due to the rapid cooling rate, while large powders contain coarse grains due to slower cooling rates during the atomization process, which is consistent with earlier reports [20,22].

Figure 2a shows the particle size analysis (PSA) of different size ranges of p-type Cu_0.07_Bi_0.5_Sb_1.5_Te_3_ powders fabricated by water atomization. Average particle sizes of about 38.84 μm, 106.09 μm, 169.50 μm, and 72.24 μm were obtained for the 32–75 μm, 75–125 μm, 125–200 μm, <200 μm powders, respectively. This PSA data is consistent with the SEM powder morphology in Figure 1. The oxygen contamination of the water atomized powders is shown in Figure 2b. It was observed that the oxygen content increased as particle size decreased, similar to gas-atomized powder [20]. The 32–75 μm powder size showed the maximum oxygen content of 668 ppm. This was mainly due to the increase in surface to volume ratio with particle size [20]. However, the <200 μm powder showed an oxygen content higher than the 75–125 μm and the 125–200 μm powders as the powder had a wide range in sizes, of about 0–200 μm. High oxygen content is mainly observed with fine powders [20]. This oxygen contamination can affect the transport behavior of carriers, as explained in further sections.

Figure 3 shows SEM fracture surface micrographs of the p-type Cu_0.07_Bi_0.5_Sb_1.5_Te_3_ samples, sintered using hot pressing. Laminar structures can be clearly observed on the fracture surface [19,20]. In addition, all specimens were well compacted, and no porosity was observed in the bulk samples. The tendency in grain size gradually increased with increasing powder size, from the 32–75 μm sample to 125–200 μm samples, which is in good agreement with the grain size behavior of powders in the cross-sectional micrographs shown in Figure 1e–g. Nevertheless, the bulk samples showed grain sizes slightly higher than the powders. The coalescence of fine grains during sintering is responsible for the grain growth in bulk samples, as the powders are sintered at 490 °C with a holding time of 30 min. Figure 3d shows the mixed grain structure due to selecting all powder sizes under 200 μm.

Figure 4 presents the XRD analysis of the (a) water atomized powders, and (b) their bulk samples. The XRD patterns of both the powders and bulk samples showed a single phase with a rhombohedral crystal structure of the R$\bar{3}$m space group, which was indexed to JCPDS: 49-1713 [19]. The Cu content in the BiSbTe alloy was too small (0.07 wt%) to be observed in the XRD analysis. Figure 4c shows the relative density and Figure 4d shows the hardness values of the p-type Cu_0.07_Bi_0.5_Sb_1.5_Te_3_ alloys. All specimens showed a relative density of over 98% of their theoretical density. This confirms that combining water atomization and hot pressing can successfully provide higher density values. The mechanical properties of the TE materials are crucial to reduce wastage during TE module preparation. The micro-Vickers hardness gradually decreased with increasing powder size, mainly due to increased grain size. The 32–75 μm specimen showed a peak hardness of 53.92 Hv, which is higher than reported for single crystal samples [23].

Figure 5a shows the electrical conductivity of the p-type Cu_0.07_Bi_0.5_Sb_1.5_Te_3_ alloys as a function of temperature. The electrical conductivity of all samples decreased quasi-linearly with increasing temperature, reflecting their highly degenerate semiconducting nature. The electrical conductivity (*σ*) can be expressed in terms of carrier concentration (*n*), and mobility (*μ*) as: *σ =neμ*, where *e* is the charge of the carrier [19]. The *n* and *μ* values were measured and are listed in Table 1. Carrier mobilities of 2.72 × 10^2^, 3.1 × 10^2^, and 3.3 × 10^2^ cm^2^/Vs were measured for the 32–75 μm, 75–125 μm, and 125–200 μm specimens, respectively. It was noted that the *μ* values gradually increased with increasing powder size, from the 32–75 μm specimen to the 125–200 μm specimen, due to increasing grain sizes, which is consistent with the earlier report [20]. It is known that carrier mobility is very sensitive to the orientation of grains, and the Bi_2_Te_3_ crystal system exhibits anisotropic electronic properties. Recently, Manzano et al. prepared highly oriented Bi_2_Te_3_ films by pulsed electrodeposition, and achieved high electronic transport properties perpendicular to the c-axis that were about 5 times greater than parallel to the c-axis [24]. This occurs as the carriers can strongly propagate (high mobility) along the specific directions of grains, which in this case were perpendicular to the c-axis. Thus, single grains or coarse grains can promote carrier transport, by reducing the scattering of carriers at grain boundaries. Note that all of our specimens were prepared perpendicular to the pressing direction. Meanwhile, the 32–75 μm specimen exhibited the lowest carrier mobility, as fine grain boundaries severely scatter carriers [19,20]. In addition, as oxygen content decreased from the 32–75 μm specimen to the 125–200 μm specimen, carrier transport improved, as the high density of the oxidation layers can severely scatter carriers, drastically decreasing carrier mobility values [20]. The carrier concentration consequently decreased as the *n* and *μ* are inversely proportional to each other. It is evident from Figure 5a that the large powder size (125–200 μm) specimen exhibited the highest electrical conductivity of 1457.33 Ω^−1^ cm^−1^ due to high mobility. The electrical conductivity values gradually decreased with decreasing powder size, due to reduced grain sizes. Fine grains can severely scatter carriers due to the high density of grain boundaries, which causes a decrease in mobility, and which can be solely responsible for the decreasing electrical conductivity.

Figure 5b shows the Seebeck coefficient (*α*) of the p-type Cu_0.07_Bi_0.5_Sb_1.5_Te_3_ alloys as a function of temperature. The positive sign confirms the p-type semiconducting nature of all specimens [19,23]. Irrespective of powder size, all the samples showed a similar trend with temperature. Notably, the *α* values of all samples increased with temperature up to 425 K, then decreased. The decreasing *α* values were mainly due to the excitation of minority carriers which contributed to the intrinsic conduction at high temperature [25]. Nevertheless, the present Cu added Bi_0.5_Sb_1.5_Te_3_ WA powder specimens exhibited a more feeble excitation of minority carriers than previously reported pure Bi-Sb-Te alloys prepared by GA + SPS, as the addition of Cu controls the intrinsic conduction [17,18]. From Figure 5b, the Seebeck coefficient values slightly increased with decreasing powder size due to the reduction in grain size. The maximum Seebeck coefficient of 208.52 μV/K at 425 K was obtained for the 32–75 μm specimen. As previously described, the 32–75 μm specimen exhibited fine grains, thereby low mobility, due to the severe scattering of carriers at the fine grain boundaries. Meanwhile, the 125–200 μm specimen showed a low Seebeck coefficient as it contained coarse grains, which have low carrier scattering. The power factor in Figure 5c shows the higher values of 4.20, 4.22 × 10^−3^ W/mK^2^ for the larger powder size samples, 125–200 μm and 75–125 μm, respectively, due to their higher electrical conductivity values. The present specimens prepared with water atomization powders revealed a higher power factor than previously reported gas atomized specimens, due to their high electrical conductivity, resulting from the 0.07 wt% Cu content in the Bi_0.5_Sb_1.5_Te_3_ matrix [19,20].

Figure 6a shows the total thermal conductivity of the p-type Cu_0.07_Bi_0.5_Sb_1.5_Te_3_ alloys as a function of temperature. The total thermal conductivity (κ) is the sum of the electronic constituent (κ*_e_*) and phonon constituent (κ_ph_) [17,18,19]. The κ*_e_* was calculated using the Lorenz number, and is shown in Figure 6b. It is evident that the total thermal conductivity gradually decreased with increasing temperature up to 400 K, and then gradually increased due to bipolar conduction behavior [18,19]. It is noteworthy that the magnitude of the intrinsic conduction is more suppressed than that of the previous report, due to the 0.07 wt% Cu content added to the Bi_0.5_Sb_1.5_Te_3_ matrix [20]. Among all the samples, the 32–75 μm specimen showed the lowest thermal conductivity, 1.17 W/mK, due to the high scattering of carriers or phonons at numerous fine grain boundaries. Meanwhile, the κ values gradually increased with increasing powder size, mainly due to the reduction in scattering of carrier/phonon with increasing grain size, consistent with earlier reports [19,20]. If the grain sizes of the specimens were further increased (beyond 1 mm), the κ values would significantly increase due to the high carriers/phonons transport. The electronic thermal conductivity (Figure 6b) increased with increasing particle size, due to the increase in electrical conductivity with increasing powder size.

The figure of merit ZT for the different investigated powder size samples are presented in Figure 6c. The 75–125 μm samples exhibited a relatively high ZT of 1.257, and the 125–200 μm showed a ZT of 1.24 at 400 K, respectively. Note that the ZT values of the 75–125 μm samples prepared by water atomization were higher than the samples prepared using gas atomization with grain sizes between 5–30 μm (ZT = 1.1) [26], a single grown crystal using the Bridgman technique (ZT = 1.0) [27], and our previous reports using gas atomization followed by spark plasma sintering (ZT = 1.12) [19]. The addition of 0.07 wt% Cu content to the Bi_0.5_Sb_1.5_Te_3_ matrix is credited for the higher values of ZT, due to high electric conductivity, compared to pure BiSbTe from the gas atomization process [20]. The high ZT of 1.257 was achieved using water atomization, a large-scale powder fabrication technique which is lower cost and easier to handle than the gas atomization process, as it does not require vacuum and inert gases. Based on these results, fabricating thermoelectric materials using water atomization and hot pressing can boost the commercial applications of thermoelectric devices, while providing low cost, less time-consuming production.

## 4. Conclusions

We fabricated a large quantity (2–3 kgs) of p-type Cu_0.07_Bi_0.5_Sb_1.5_Te_3_ alloy powders using an inexpensive and rapid solidification process, water atomization, and sintering by hot pressing. The 32–75 μm powders contained high oxygen contamination due to their high surface to volume ratio. It was determined that the grain size gradually increased with increasing particle size, which severely impacted the mechanical and transport properties of the thermoelectric materials. The 32–75 μm specimen displayed the highest Vickers hardness of 53.92 Hv compared to other specimens, due to its fine grain structure. A noticeable increase in carrier mobility of about 330 cm^2^ V^−1^ s^−1^ was observed for the 125–200 μm specimen. This was also responsible for the increase in electrical conductivity, to 1457.33 Ω^−1^ cm^−1^, reflecting the higher carrier transport due to coarse grain size. The peak Seebeck coefficient of 208.52 μV/K as well as the lowest thermal conductivity of 1.17 W/m-K were exhibited by the 32–75 μm specimen, due to the severe scattering of carriers (low carrier mobility) and phonons at the fine grain boundaries. The maximum figure of merit, ZT of 1.257 was exhibited by the 75–125 μm powder specimen, due to the high electrical conductivity values.

## Figures and Tables

**Figure 1 materials-14-04870-f001:**
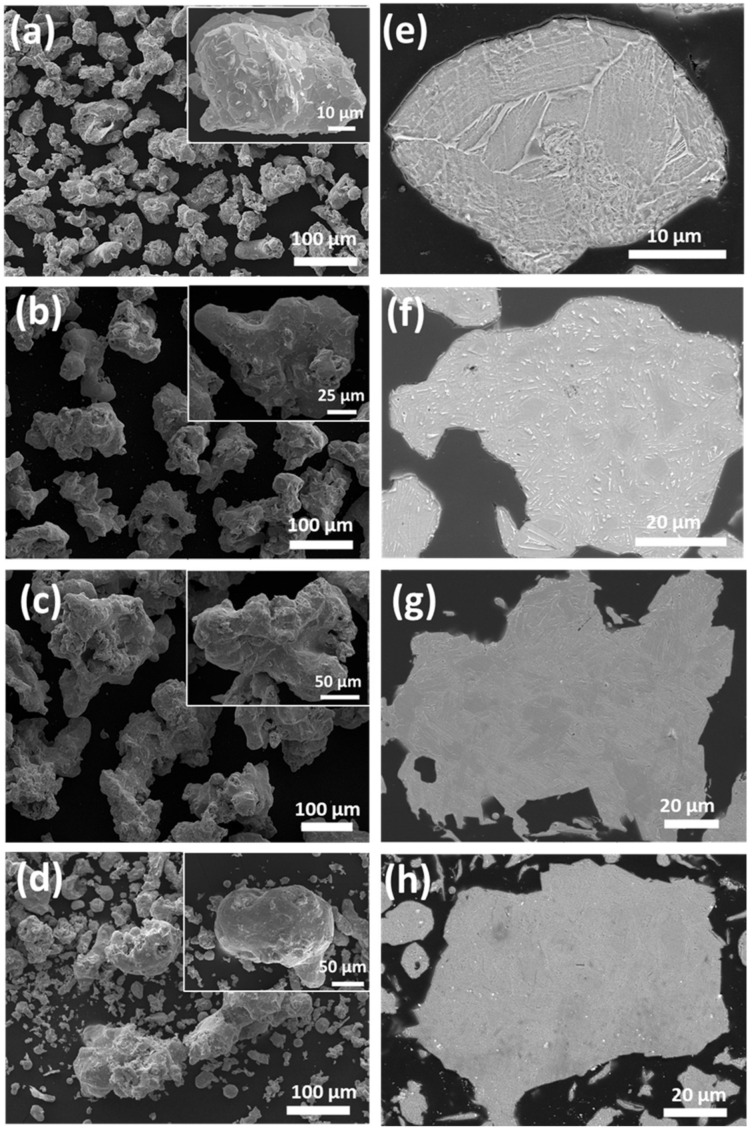
SEM powder morphology of (**a**) 32–75 μm, (**b**) 75–125 μm, (**c**) 125–200 μm, and (**d**) <200 μm size powders. The inset images are highly magnified. SEM powder cross-sectional images of (**e**) 32–75 μm, (**f**) 75–125 μm, (**g**) 125–200 μm, and (**h**) <200 μm. The insets show their corresponding high magnification images, respectively.

**Figure 2 materials-14-04870-f002:**
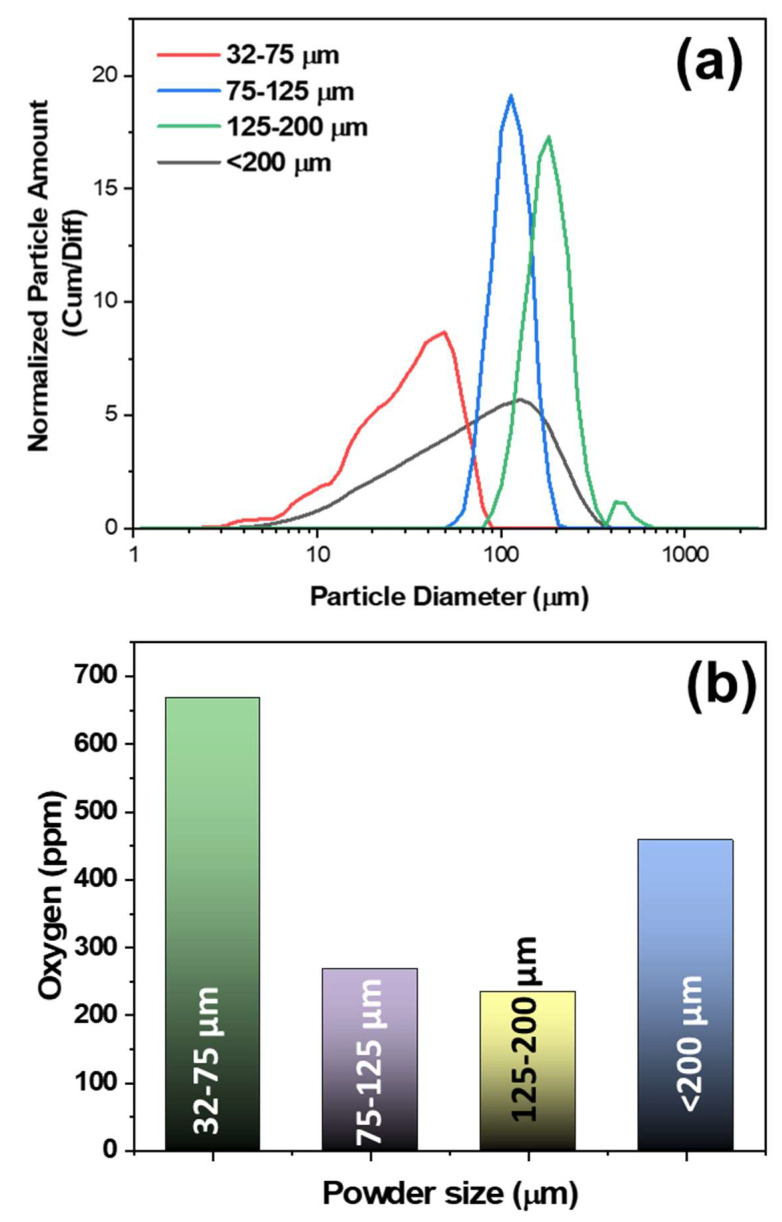
(**a**) Powder size analysis, and (**b**) oxygen contamination analysis of p-type Cu_0.07_Bi_0.5_Sb_1.5_Te_3_ water atomized powders. The small powder range (32–75 μm) had the highest content of oxygen contamination due to the high surface to volume ratio.

**Figure 3 materials-14-04870-f003:**
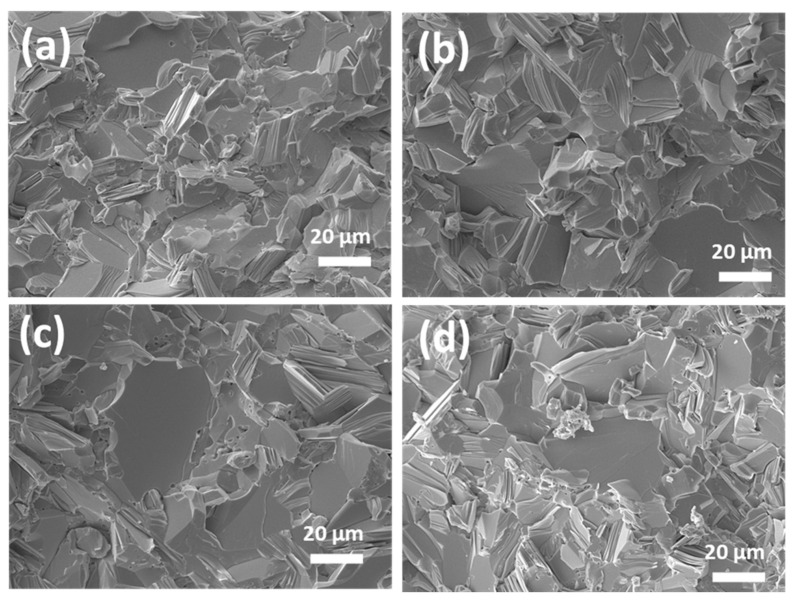
SEM bulk fracture surface analysis of different powder ranges of p-type Cu_0.07_Bi_0.5_Sb_1.5_Te_3_ bulk specimens (**a**) 32–75 μm, (**b**) 75–125 μm, (**c**) 125–200 μm, and (**d**) <200 μm samples.

**Figure 4 materials-14-04870-f004:**
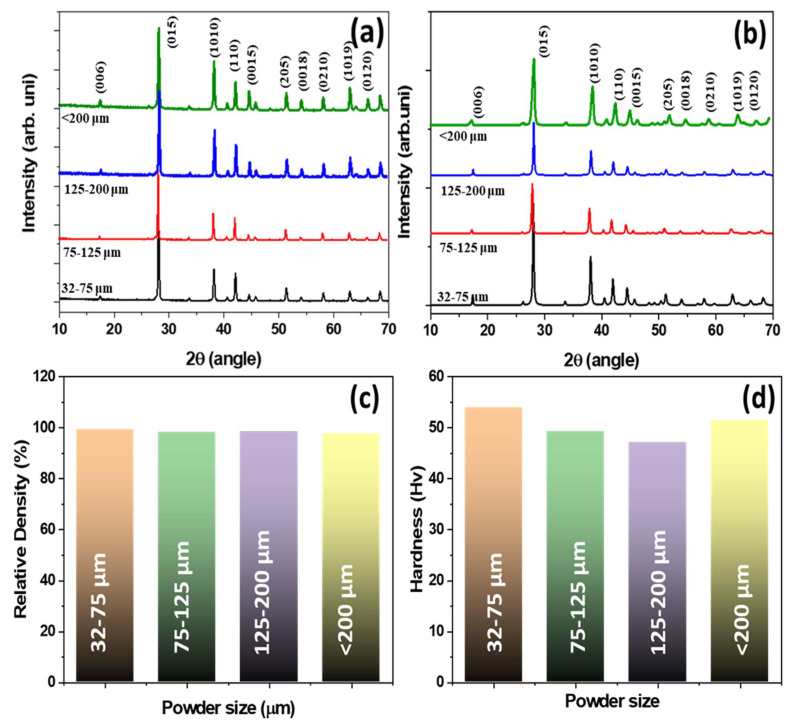
XRD analysis of different sized (**a**) powders and (**b**) bulk specimens of p-type Cu_0.07_Bi_0.5_Sb_1.5_Te_3_ alloys. (**c**) relative density measurement and (**d**) micro-Vickers hardness analysis of p-type Cu_0.07_Bi_0.5_Sb_1.5_Te_3_ bulk specimens. All of the specimen XRD peaks confirmed the formation of Bi_0.5_Sb_1.5_Te_3_ single phase (JCPSD#49-1713). The 32–75 μm bulk specimen had higher hardness values than other samples, due to its fine grain microstructure.

**Figure 5 materials-14-04870-f005:**
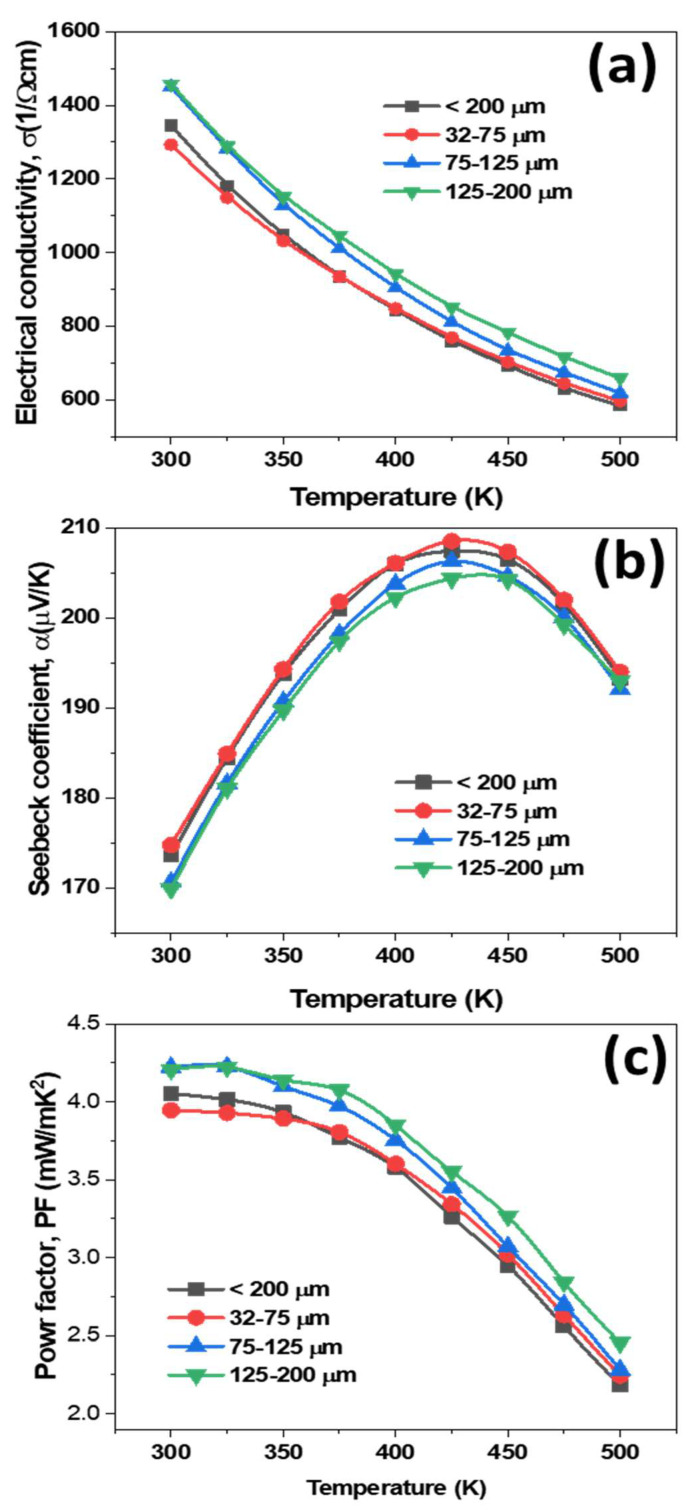
Temperature dependence of (**a**) electrical conductivity, (**b**) Seebeck coefficient, and (**c**) power factor for different water atomized powder sizes of p-type Cu_0.07_Bi_0.5_Sb_1.5_Te_3_ bulk specimens.

**Figure 6 materials-14-04870-f006:**
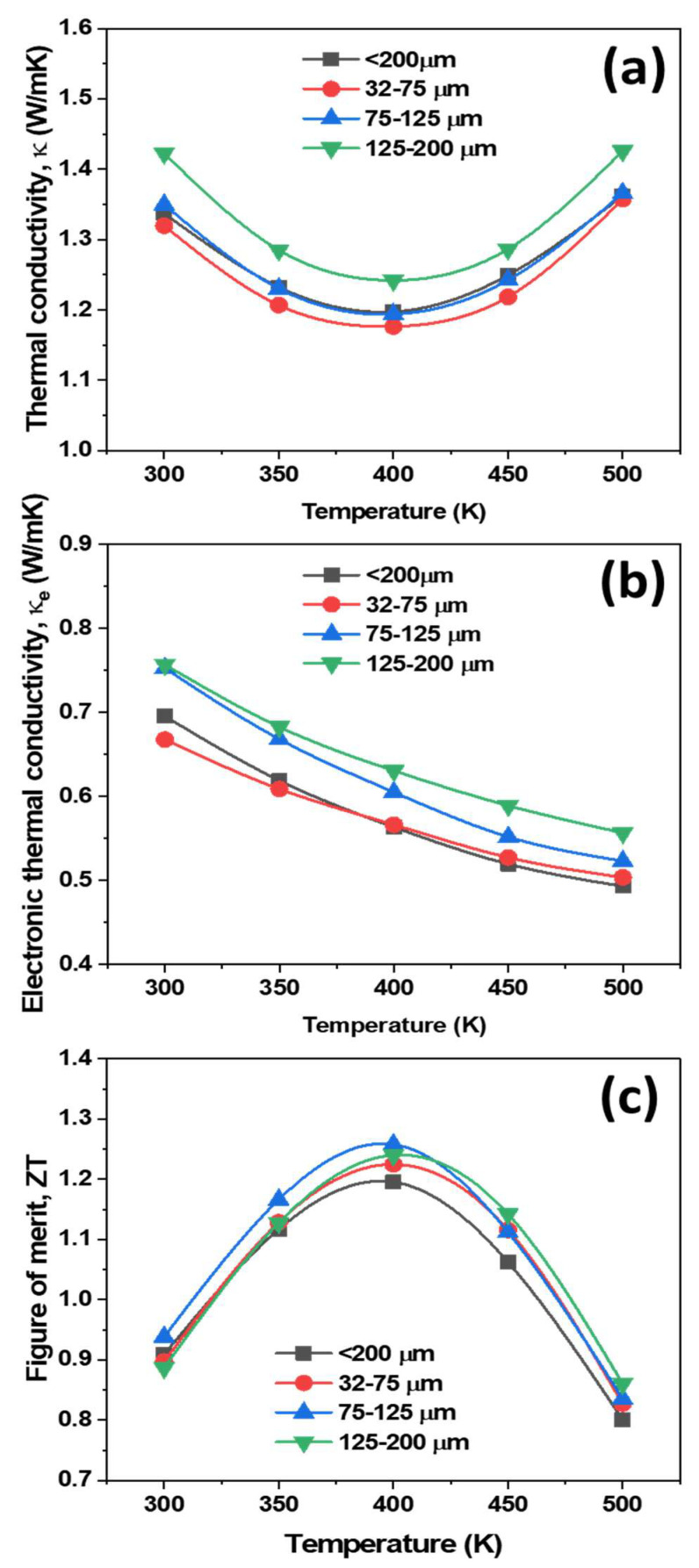
Temperature dependence of (**a**) total thermal conductivity, (**b**) electronic thermal conductivity, and (**c**) figure of merit for different water atomized powder sizes of p-type Cu_0.07_Bi_0.5_Sb_1.5_Te_3_ bulk specimens.

**Table 1 materials-14-04870-t001:** Room temperature carrier mobility, and concentration of p-type Cu_0.07_Bi_0.5_Sb_1.5_Te_3_ bulk specimens.

Sample Name	Carrier Mobility (*μ*) (10^2^ cm^2^ V^−1^ s^−1^)	Carrier Concentration (*n*) (10^19^ cm^−3^)
32–75 μm	2.72	2.83
75–125 μm	3.1	2.69
125–200 μm	3.3	2.49
<200 μm	3.18	2.86

## Data Availability

The data presented in this study are available on request from the corresponding author.

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
