# Peer review of "Development of High-Performance Thermoelectric Materials by Microstructure Control of P-Type BiSbTe Based Alloys Fabricated by Water Atomization"

_materials, 2021, doi:10.3390/ma14174870_

Round 1

Reviewer 1 Report

This paper reports the development of high-performance thermoelectric materials through microstructure controlling by optimizing initial powder size of p-type BiSbTe based alloys fabricated by water atomization. In general, the manuscript is well-organized, logically laid out and the experimental approach is technically sound. This paper is possibly publishable. For improving a manuscript, it is advisable to address the following comments:

1.    Please explain why the alloy powder in size of > 200 um was not investigated.
2.    In Fig.5(b), it was found that the Seebeck coefficient of all alloys increased but then decreased during the increasing temperature. Please explain it.
3.    In Fig.6(a), it was seen that the thermal conductivity of all alloys decreased but then increased during the increasing temperature. Please explain it.
4.    How about the performance of your thermoelectric alloy materials in this method, compared with others reported in the literature?   

Reviewer 2 Report

Please find my comments attached.

Round 2

Reviewer 2 Report

The authors have made reasonable changes in light of the comments from the reviewer, however, my concern remains at large related to English writing style. The author's notes (cover letter) as well as the manuscript, still contains many punctuation, grammatical and sentence structure mistakes.

For example:

 Meanwhile, Cu has chosen as a dopant in BiSbTe system to suppress of intrinsic conduction at elevated temperatures. 

and

 The fabricated powders have wide range of particle sizes due to the different in cooling rate during the process 

etc.

These mistakes are numerous throughout and matter of seriousness. This manuscript should not pass to publication with such trivial mistakes and crude writing style.

Also when mentioning the instruments in experimental section on which the analytical task was conducted, please specify its details and make in full.

Following these correction, I can recommend this manuscript for publication at MDPI Materials.

Author Response

Thank you for the reviewer suggestions to our manuscript. We have modified the manuscript with native English speaker through English Language Editing center. [Please see the attachment for English correction certificate]

In addition, we have added the instrument full details with corresponding company names in the experimental part, and added the specimen dimensions for the thermoelectric measurements.

[Please see corrections in the Page#5&Lines#16-23; and Page#6 & lines#1-7]
